# Searching for Higgs Boson Decay Modes with Deep Learning

**Peter Sadowski**
Department of Computer Science
University of California, Irvine
Irvine, CA 92617
peter.j.sadowski@uci.edu

**Pierre Baldi**
Department of Computer Science
University of California, Irvine
Irvine, CA 92617
pfbaldi@ics.uci.edu

**Daniel Whiteson**
Department of Physics and Astronomy
University of California, Irvine
Irvine, CA 92617 Address
daniel@uci.edu

## Abstract

Particle colliders enable us to probe the fundamental nature of matter by observing exotic particles produced by high-energy collisions. Because the experimental measurements from these collisions are necessarily incomplete and imprecise, machine learning algorithms play a major role in the analysis of experimental data. The high-energy physics community typically relies on standardized machine learning software packages for this analysis, and devotes substantial effort towards improving statistical power by hand-crafting high-level features derived from the raw collider measurements. In this paper, we train artificial neural networks to detect the decay of the Higgs boson to tau leptons on a dataset of 82 million simulated collision events. We demonstrate that *deep* neural network architectures are particularly well-suited for this task with the ability to *automatically* discover high-level features from the data and increase discovery significance.

## 1 Introduction

The Higgs boson was observed for the first time in 2011-2012 and will be the central object of study when the Large Hadron Collider (LHC) comes back online to collect new data in 2015. The observation of the Higgs boson in $ZZ, \gamma\gamma$, and $WW$ decay modes at the LHC confirms its role in electroweak symmetry-breaking [1, 2]. However, to establish that it also provides the interaction which gives mass to the fundamental fermions, it must be demonstrated that the Higgs boson couples to fermions through direct decay modes. Of the available modes, the most promising is the decay to a pair of tau leptons ($\tau$), which balances a modest branching ratio with manageable backgrounds. From the measurements collected in 2011-2012, the LHC collaborations report data consistent with $h \to \tau\tau$ decays, but without statistical power to cross the $5\sigma$ threshold, the standard for claims of discovery in high-energy physics.

Machine learning plays a major role in processing data at particle colliders. This occurs at two levels: the *online* filtering of streaming detector measurements, and the *offline* analysis of data once it has been recorded [3], which is the focus of this work. Machine learning classifiers learn to distinguish between different types of collision events by training on simulated data from sophisticated Monte-Carlo programs. Single-hidden-layer, *shallow* neural networks are one of the primary techniques used for this analysis, and standardized implementations are included in the prevalent multi-variate

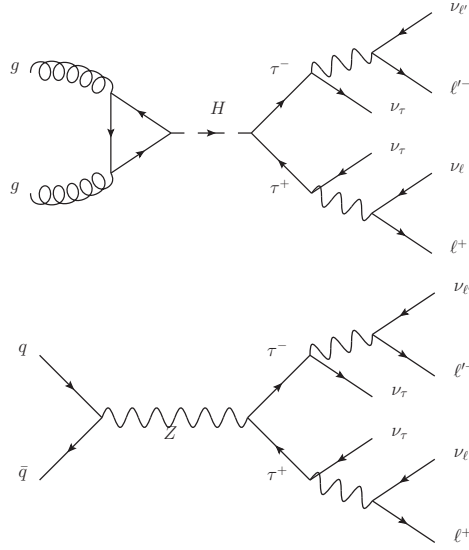

Figure 1: Diagrams for the signal $gg \to h \to \tau\tau \to \ell^- \nu\nu\ell^+\nu\nu$ and the dominant background $q\bar{q} \to Z \to \tau\tau \to \ell^- \nu\nu\ell^+\nu\nu$.

analysis software tools used by physicists. Efforts to increase statistical power tend to focus on developing new features for use with the existing machine learning classifiers — these high-level features are non-linear functions of the low-level measurements, derived using knowledge of the underlying physical processes.

However, the abundance of labeled simulation training data and the complex underlying structure make this an ideal application for large, deep neural networks. In this work, we show how deep neural networks can simplify and improve the analysis of high-energy physics data by *automatically* learning high-level features from the data. We begin by describing the nature of the data and explaining the difference between the low-level and high-level features used by physicists. Then we demonstrate that deep neural networks increase the statistical power of this analysis *even without the help of manually-derived high-level features*.

## 2   Data

Collisions of protons at the LHC annhiliate the proton constituents, quarks and gluons. In a small fraction of collisions, a new heavy state of matter is formed, such as a Higgs or $Z$ boson. Such states are very unstable and decay rapidly and successively into lighter particles until stable particles are produced. In the case of Higgs boson production, the process is:

$$gg \to H \to \tau^+\tau^- \tag{1}$$

followed by the subsequent decay of the $\tau$ leptons into lighter leptons ($e$ and $\mu$) and pairs of neutrinos ($\nu$), see Fig. 1.

The point of collision is surrounded by concentric layers of detectors that measure the momentum and direction of the final stable particles. The intermediate states are not observable, such that two different processes with the same set of final stable particles can be difficult to distinguish. For example, Figure 1 shows how the process $q\bar{q} \to Z \to \tau^+\tau^-$ yields the identical list of particles as a process that produces the Higgs boson.

The primary approach to distinguish between two processes with identical final state particles is via the momentum and direction of the particles, which contain information about the identity of the intermediate state. With perfect measurement resolution and complete information of final state

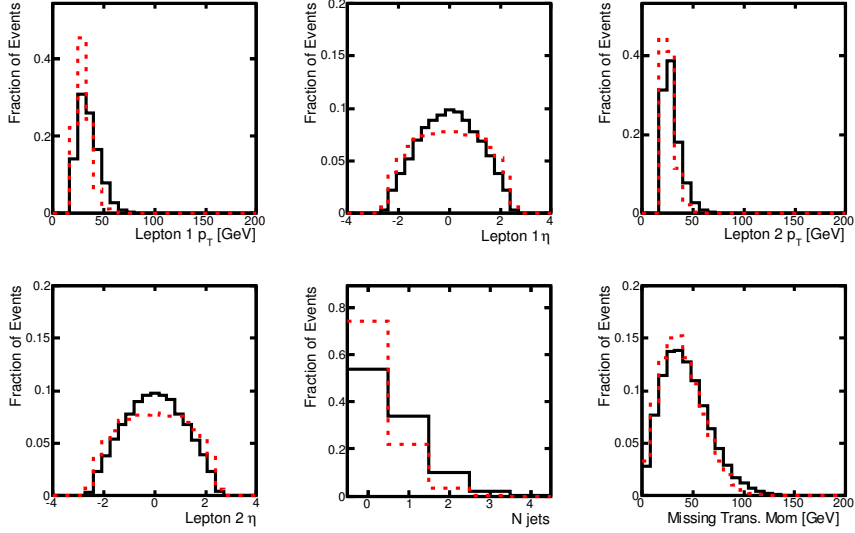

Figure 2: Low-level input features from basic kinematic quantities in $\ell\ell + p\!\!\!/_T$ events for simulated signal (black) and background (red) benchmark events. Shown are the distributions of transverse momenta ($p_T$) of each observed particle as well as the imbalance of momentum in the final state. Momentum angular information for each observed particle is also available to the network, but is not shown, as the one-dimensional projections have little information.

particles $B$ and $C$, we could calculate the invariant mass of the short-lived intermediate state $A$ in the process $A \to B + C$, via:

$$m_A^2 = m_{B+C}^2 = (E_B + E_C)^2 - |(p_B + p_C)|^2 \tag{2}$$

However, finite measurement resolution and escaping neutrinos (which are invisible to the detectors) make it impossible to calculate the intermediate state mass precisely. Instead, the momentum and direction of the final state particles are studied. This is done using simulated collisions from sophisticated Monte Carlo programs [4, 5, 6] that have been carefully tuned to provide highly faithful descriptions of the collider data. Machine learning classifiers are trained on the simulated data to recognize small differences in these processes, then the trained classifiers are used to analyze the experimental data.

## 2.1 Low-level features

There are ten low-level features that comprise the essential measurements provided by the detectors:

- The three-dimensional momenta, $p$, of the charged leptons;
- The imbalance of momentum ($p\!\!\!/_T$) in the final state transverse to the beam direction, due to unobserved or mismeasured particles;
- The number and momenta of particle 'jets' due to radiation of gluons or quarks.

Distributions of these features are given in Fig. 2.

## 2.2 High-level features

There is a vigorous effort in the physics community to construct non-linear combinations of these low-level features that improve discrimination between Higgs-boson production and $Z$-boson production. High-level features that have been considered include:

- Axial missing momentum, $p\!\!\!/_T \cdot p_{\ell^+\ell^-}$;

- Scalar sum of the observed momenta, $|p_{\ell^+}| + |p_{\ell^-}| + |\not{p}_T| + \sum_i |p_{\mathrm{jet}_i}|$;

- Relative missing momentum, $\not{p}_T$ if $\Delta\phi(p,\not{p}_T) \geq \pi/2$, and $\not{p}_T \times \sin(\Delta\phi(p,\not{p}_T)$ if $\Delta\phi(p,\not{p}_T) < \pi/2$, where $p$ is the momentum of any charged lepton or jet;

- Difference in lepton azimuthal angles, $\Delta\phi(\ell^+,\ell^-)$;

- Difference in lepton polar angles, $\Delta\eta(\ell^+,\ell^-)$;

- Angular distance between leptons, $\Delta R = \sqrt{(\Delta\eta)^2 + (\Delta\phi)^2}$;

- Invariant mass of the two leptons, $m_{\ell^+\ell^-}$;

- Missing mass, $m_{\mathrm{MMC}}$ [7];

- Sphericity and transverse sphericity;

- Invariant mass of all visible objects (leptons and jets).

Distributions of these features are given in Fig. 3.

# 3 Methods

## 3.1 Current approach

Standard machine learning techniques in high-energy physics include methods such as boosted decision trees and single-layer feed-forward neural networks. The TMVA package [8] contains a standardized implementation of these techniques that is widely-used by physicists. However, we have found that our own hyperparameter-optimized, single-layer neural networks perform better than the TMVA implementations. Therefore, we use our own hyperparameter-optimized shallow neural networks trained on fast graphics processors as a benchmark for comparison.

## 3.2 Deep learning

Deep neural networks can automatically learn a complex hierarchy of non-linear features from data. Training deep networks often requires additional computation and a careful selection of hyperparameters, but these difficulties have diminished substantially with the advent of inexpensive graphics processing hardware. We demonstrate here that deep neural networks provide a practical tool for learning deep feature hierarchies and improving classifier accuracy while reducing the need for physicists to carefully derive new features by hand. Many exploratory experiments were carried with different architectures, training protocols, and hyperparameter optimization strategies. Some of these experiments are still ongoing and, for conciseness, we report only the main results obtained so far.

## 3.3 Hyperparameter optimization

Hyperparameters were optimized separately for shallow and deep neural networks. Shallow network hyperparameters were chosen from combinations of the parameters listed in Table 1, while deep network hyperparameters were chosen from combinations of those listed in Table 2. These were selected based on classification performance (cross-entropy error) on the validation set, using the full set of available features: 10 low-level features plus 15 high-level features. The best architectures were the largest ones: a deep network with 300 hidden units in each of five hidden layers and an initial learning rate of 0.03, and a shallow network with 15000 hidden units and an initial learning rate of 0.01. These neural networks have approximately the same number of tunable parameters, with 369,301 parameters in the deep network and 405,001 parameters in the shallow network.

Table 1: Hyperparameter options for shallow networks.

| Hyperparameter | Options |
|---|---|
| Hidden units | 100, 300, 1000, 15000 |
| Initial learning rate | 0.03, 0.01, 0.003, 0.001 |

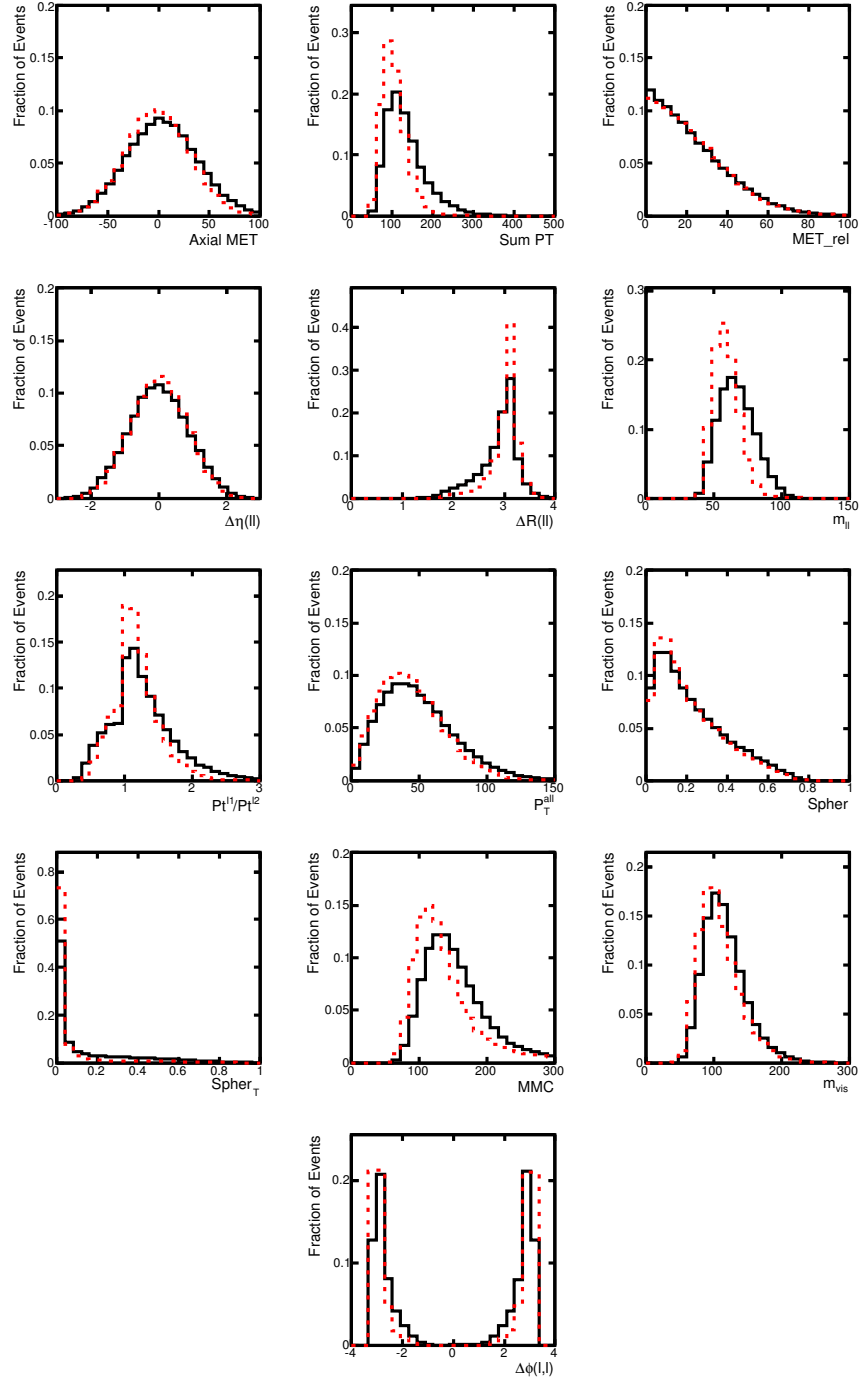

Figure 3: Distribution of high-level input features from invariant mass calculations in $\ell\nu jjb\bar{b}$ events for simulated signal (black) and background (red) events.

## 3.4 Training details

The problem is a basic classification task with two classes. The data set is balanced and contains 82 million examples. A validation set of 1 million examples was randomly set aside for tuning the hyperparameters. Different cross validation strategies were used with little influence on the results reported since these are obtained in a regime far away from overfitting.

Table 2: Hyperparameter options for deep networks.

| Hyperparameter | Options |
|---|---|
| Number of layers | 3,4,5,6 |
| Hidden units per layer | 100, 300 |
| Initial learning rate | 0.03, 0.01, 0.003 |

The following neural network hyperparameters were predetermined without optimization. The *tanh* activation function was used for all hidden units, while the the *logistic* function was used for the output. Weights were initialized from a normal distribution with zero mean and standard deviation 0.1 in the first layer, 0.001 in the output layer, and $\frac{1}{\sqrt{k}}$ for all other hidden layers, where $k$ was the number of units in the previous layer. Gradient computations were made on mini-batches of size 100. A momentum term increased linearly over the first 25 epochs from 0.5 to 0.99, then remained constant. The learning rate decayed by a factor of 1.0000002 every batch update until it reached a minimum of $10^{-6}$. All networks were trained for 50 epochs.

Computations were performed using machines with 16 Intel Xeon cores, an NVIDIA Tesla C2070 graphics processor, and 64 GB memory. Training was performed using the Theano and Pylearn2 software libraries [9, 10].

## 4 Results

The performance of each neural network architecture in terms of the Area Under the signal-rejection Curve (AUC) is shown in Table 3. As expected, the shallow neural networks (one hidden layer) perform better with the high-level features than the low-level features alone; the high-level features were specifically designed to discriminate between the two hypotheses. However, this difference disappears in deep neural networks, and in fact performance is *better* with the 10 low-level features than with the 15 high-level features alone. This, along with the fact that the complete set of features always performs best, suggests that there is information in the low-level measurements that is not captured by the high-level features, and that the deep networks are exploiting this information.

Table 3: Comparison of performance for neural network architectures: shallow neural networks (NN), and deep neural networks (DN) with different numbers of hidden units and layers. Each network architecture was trained on three sets of input features: low-level features, high-level features, and the complete set of features. The table displays the test set AUC and the expected significance of a discovery (in units of Gaussian $\sigma$) for 100 signal events and 5000 background events with a 5% relative uncertainty.

| | AUC | | |
|---|---|---|---|
| Technique | Low-level | High-level | Complete |
| NN 300 | 0.788 | 0.792 | 0.798 |
| NN 1000 | 0.788 | 0.792 | 0.798 |
| NN 15000 | 0.788 | 0.792 | 0.798 |
| DN 3-layer | 0.796 | 0.794 | 0.801 |
| DN 4-layer | 0.797 | 0.797 | 0.802 |
| DN 5-layer | 0.798 | 0.798 | 0.803 |
| DN 6-layer | 0.799 | 0.797 | 0.803 |
| | Discovery significance | | |
| Technique | Low-level | High-level | Complete |
| NN 15000 | $1.7\sigma$ | $2.0\sigma$ | $2.0\sigma$ |
| DN 6-layer | $2.1\sigma$ | $2.2\sigma$ | $2.2\sigma$ |

The best networks are trained with the complete set of features, which provides both the raw measurements and the physicist's domain knowledge. Figure 4 plots the empirical distribution of predictions (neural network output) for the test samples from each class, and shows how both the shallow and deep networks trained on the complete feature set are more confident about their correct predictions.

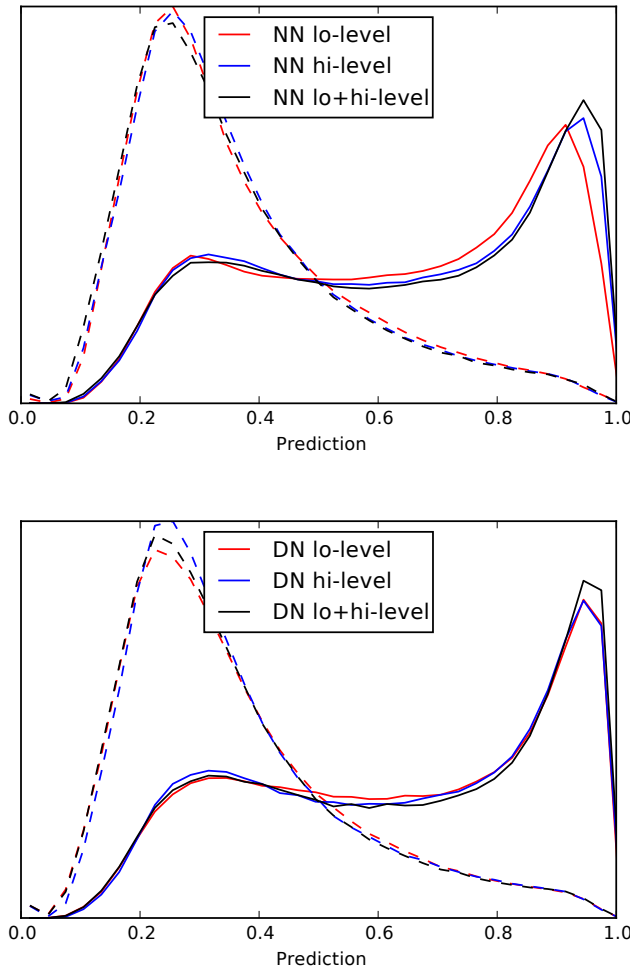

Figure 4: Empirical distribution of predictions for signal events (solid) and background events (dashed) from the test set.

Figure 5 shows how the AUC translates into discovery significance [11]. On this metric too, the six-layer deep network trained on the low-level features outperforms the best shallow network (15000 hidden units) trained with the best feature set.

## 5 Discussion

While deep learning has led to significant advances in computer vision, speech, and natural language processing, it is clearly useful for a wide range of applications, including a host of applications in the natural sciences. The problems in high-energy physics are particularly suitable for deep learning, having large data sets with complex underlying structure. Our results show that deep neural networks provide a powerful and practical approach to analyzing particle collider data, and that the high-level features learned from the data by deep neural networks increase the statistical power *more* than the common high-level features handcrafted by the physicists. While the improvements may seem small, they are very significant, especially when considering the billion-dollar cost of accelerator experiments.

These preliminary experiments demonstrate the advantages of deep neural networks, but we have not yet pushed the limits of what deep learning can do for this application. The deep architectures in this work have less than 500,000 parameters and have not even begun to overfit the training data.

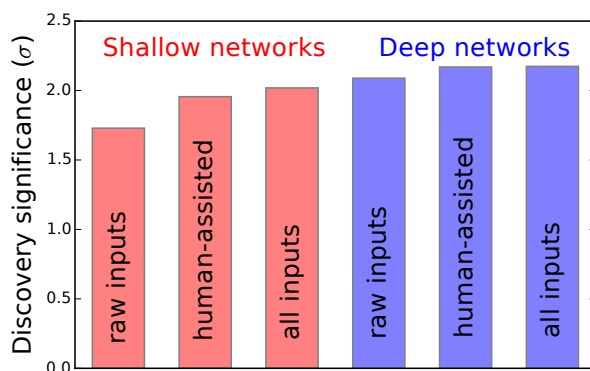

Figure 5: Comparison of discovery significance for the traditional learning method (left) and the deep learning method (right) using the low-level features, the high-level features, and the complete set of features.

Experiments with larger architectures, including ensembles, with a variety of shapes and neuron types, are currently in progress.

Since the high-level features are derived from the low-level features, it is interesting to note that one could train a regression neural network to learn this relationship. Such a network would then be able to predict the physicist-derived features from the low-level measurements. Some of these high-level features may be more difficult to compute than others, requiring neural networks of a particular size and depth, and it would be interesting to analyze the complexity of the high-level features in this way. We are in the process of training such regression networks which could then be incorporated into a larger prediction architecture, either by freezing their weights, or by allowing them to learn further. In combination, these deep learning approaches should yield a system ready to sift through the new Large Hadron Collider data in 2015.

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
