[Reviews · NeurIPS 2014]

Submitted by Assigned_Reviewer_42

The paper describes an application of deep neural networks to offline high energy physics particle collider event detection. The data used for training and testing were generated by simulating collisions. The paper describes experiments using a small number of features that correspond to the actual detector measurements as well as engineered features previously proposed for this task.

The paper describes a reasonable machine learning approach to an important problem.
As an application paper, in order to add substantial value to the conference, the paper needs to provide a practically important improvement over previous approaches to the problem. Furthermore, the choices made when applying the model to the problem need to be clearly explained, justified, and ideally supported empirically. How are the discovery significance numbers in table 3 calculated and what evidence is there that the improvement over the baseline will be exciting to the physicists who would make use of the classifier? Any citation or argument understandable to people outside HEP that supports the usefulness of the improvement would strengthen the paper dramatically.

The caption for figure 2 and section 2.1 need clarification. By my count, there are 6 momenta inputs for the charged leptons and 1 transverse momentum imbalance input. However, it isn't completely clear how the jets are dealt with. Are there 3 more jet momenta inputs that hold the sum of the momenta of all jets? And then one more input holding the number of jets? Presumably the momentum angular information mentioned in the caption for figure 2 is not used. How exactly are the inputs encoded?

The first sentence of section 2.2 would be improved by some sort of citation to an example of something in the physics literature that deals with some of these high level features.

If at all possible, the exact dataset used should be released publicly.

The empirical investigation of different neural net architectures and metaparameter settings feels somewhat preliminary and unfinished, especially since the largest networks got the best results and there is no obvious reason why larger networks couldn't be tried relatively quickly with a couple of the machines used. Were any other momentum schedules tried other than ramping up to 0.99? Or a learning rate annealing schedule based on the training progress? How about rectified linear units? The text mentions that there is no overfitting, but to be maximally clear it should mention somewhere that there is no gap between training set and validation set error or provide a training curve. A training curve with training set and validation set progress as a function of epoch number would help the reader judge whether stopping after 50 epochs was reasonable.

How large is the test set? Adding a count scale to the ordinate of the plots in figure 4 would have partially answered this question, but it would probably be better just to say how large the test set is.

Figure 5 is not particularly valuable and could be eliminated if more space is needed.

In general, in an application paper that uses a deep neural net classifier, there should be a strenuous effort to squeeze the most performance possible out of the neural net, possibly by using Bayesian optimization to set the metaparameters.
Summary: The paper describes a reasonable deep neural net approach to an interesting application, but the experimental evaluation feels very preliminary.

Submitted by Assigned_Reviewer_44

This paper describes the use of deep learning for finding Higgs boson decay modes in accelerator experimental data, and how the approach outperforms handcrafted features.

This is a very straightforward paper, extremely well written and clear. The approach taken is sound, and the experimental results convincing. The result is significant for the scientific community at large, highlighting the potential of deep learning techniques in areas beyond traditional human perception and NLP tasks. It's very useful to highlight that the approach is able to learn and complement handcrafted features.

Some comments to the authors:
1- Figure 1 legend 'h' should be 'H'
2- Figure 4 is not readable when printed in black and white (hence unlikely to be accessible to colorblind readers).

This reviewer is curious about how this work relates to the ongoing machine learning competition here: https://www.kaggle.com/c/higgs-boson
I would also suggest trying much deeper networks with rectified linear non-linearities in the hidden layers (they are much more robust to adding depth than tanh) and introducing dropout when you go to much larger networks.
Summary: Very good, very well written application paper. The fact that it suggests deep learning could be relevant to a whole new set of applications is very important and has a very strong potential for impact.

Submitted by Assigned_Reviewer_47

SUMMARY:
This paper discusses the use of deep neural networks in a novel application: distinguishing Higgs boson decay to tau particles from the much more frequent background event of Z boson decay to tau particles. The authors compare results of shallow networks versus deep networks on the task at hand and demonstrate better performance for deep networks. They also use the scores of the shallow versus deep networks to analyse the role of basic vs. handcrafted features.

GENERAL OPINION:
This is a very straightforward paper that applies an existing technique on a novel, relevant and interesting application. The technical quality and clarity are very sound. Even though there is no real novelty on the side of machine learning research itself, in my opinion the introduction of a new application domain for deep learning in an extremely relevant scientific field in itself warrants acceptance.

One caveat is that this reviewer has a background in physics, and is hence quite familiar with (and enthusiastic about) the concepts explained in the introduction. Whereas I found this explanation very clear, I cannot really speak for the rest of the NIPS community.

DETAILED COMMENTS:

- If possible, it might be informative to provide a depiction that illustrates the raw data one gets from a typical collision event (an illustration of the detected particles emerging from a single collision). This might give a better feel of the kind of data that is dealt with.

- In line 52 the authors state that there is a complex underlying structure to the data. Could this be explained in more detail? Are there compelling physical reasons to assume this?

- Table 3 provides discovery significances for the experiments. Can they be used to make quantitative predictions on real physical experiments? For instance: could you claim that fewer data would need to be collected to reach the 5 sigma significance, and if so, what would be the expected reduction in time/effort compared to using shallow networks?

- Line 267: "The dataset is balanced". I found this somewhat surprising, given the fact that the Higgs-boson should only appear in a very small number of cases. What is the motivation for this? Is the physical imbalance of the data taken into account when computing significance levels? It would be informative to explain at least roughly how the significance levels are computed.

Summary: Overall a very straightforward and nice paper. Some tweaking to the explanations would benefit the general non-physicist reading community.
Author Feedback
Author rebuttal: The authors appreciate the helpful comments of the anonymous reviewers and will incorporate their suggestions into the revised version of the manuscript. We will of course fix all the minor details and specific comments, such as improving the captions of figure 2 or including additional background references, and make the data publicly available upon publication. We are continuing to run experiments on the data, including Bayesian methods for hyper-parameter optimization, and the latest and best results will be included in the final version of the paper, if accepted.

The expected discovery significance was calculated by averaging the discovery significance computed from randomly-selected, unbalanced subsets of the test set, with each subset consisting of 100 signal events and 5000 background events. This step was performed according to standard statistical methods (cited in the paper) that are broadly accepted in the High-Energy Physics (HEP) community. The discovery significance corresponds to the probability that the background hypothesis (no Higgs boson) would fluctuate to give a more signal-like result than that observed in the test set. The calculation was done using the tools developed by the Large Hadron Collider (HLC) collaborations precisely for this purpose. A balanced training set was used to speed up neural network training. The AUC metric was also computed on the balanced test set consisting of one million examples.

It is essential to note that colliders are billion dollars instruments capable of producing petabytes of raw data per second. Given the magnitude of what is at stake, both in terms of investments and fundamental discoveries, the HEP community is very receptive to even small improvements in this area. Now that the Higgs boson has been discovered, the most pressing open question is whether it is responsible for the masses of the fundamental fermions, the particles that constitute all matter. If this is the case, it will also decay into these fermions, and the LHC collaborations are currently in a neck-and-neck race to make the first statistically significant observation of such decays. The detection improvements reported in the submitted version are already very significant and, as mentioned, we are running additional experiments to further improve these results.

The Kaggle competition, which was started after this work, tackles the same problem addressed by this work -- to detect the tau-tau decay mode of the Higgs boson -- albeit with some significant differences in the features and the data-producing simulations. However, the Kaggle dataset contains only a quarter of a million training examples; thus a major challenge of the competition is to avoid overfitting (the rules forbid the use of additional data). Our full dataset consists of 100 million training examples, and thus our chance of overfitting is reduced almost to zero. Since an arbitrary number of training examples can be produced through simulations, we believe that the limited training data used in the Kaggle competition actually creates an artificial bottleneck and detracts from its purported goal of finding the best approaches to this important detection problem.